

**Direct observations of organic aerosols in common**
**wintertime hazes in North China: insights into their size,**
**shape, mixing state, and source**
**S. R. Chen[1], L. Xu[1], Y. X. Zhang[1], B. Chen[1], X. F. Wang[1], X. Y. Zhang[2], M.**
**Zheng[3], J. M. Chen[1], W. X. Wang[1], Y. L. Sun[4], P. Q. Fu[4], Z. F. Wang[4], W. J. Li[1,*]**
[1] Environment Research Institute, Shandong University, Jinan, Shandong 250100,
China
[2] Key Laboratory of Atmospheric Chemistry, Chinese Academy of Meteorological
Sciences, Beijing 100081, China
[3] State Key Joint Laboratory of Environmental Simulation and Pollution Control,
College of Environmental Sciences and Engineering, Peking University, Beijing
100871, China
[4] State Key Laboratory of Atmospheric Boundary Layer Physics and Atmospheric
Chemistry, Institute of Atmospheric Physics, Chinese Academy of Sciences, Beijing,
China
[*]Corresponding Email: liweijun@sdu.edu.cn (W. J. Li)





**Abstract**
Many studies have focused on the physicochemical properties of aerosol particles in
unusually severe haze episodes instead of the more freqent and less severe hazes.
Consistent with this lack of attention, the morphology and mixing state of organic
matter (OM) particles in the frequent light and moderate (L&M) hazes in winter in
North China Plain (NCP) have not been examined, even though OM dominates these
fine particles. In the present work, morphology, mixing state, and size of organic
aerosols in the L&M hazes were systematically characterized using transmission
electron microscopy coupled with energy-dispersive X-ray spectroscopy, atomic force
microscopy, and nanoscale secondary ion mass spectrometer, with the comparisons
among an urban site (Jinan, S1), a mountain site (Tai, S2), and a background island
site (Changdao, S3) in the same hazes. Based on their morphology, the OM particles
were divided into six different types: spherical (type 1), near-spherical (type 2),
irregular (type 3), domelike (type 4), dispersed-OM (type 5), and OM-coating (type 6).
In the three sampling sites, type 1-3 of OM particles were most abundant in the L&M
hazes and most of them were internally mixed with non-OM particles. The abundant
near-spherical OM particles with higher sphericity and lower aspect ratio indicate that
these primary OM particles formed in cooling, polluted plumes from coal combustion
and biomass burning. Based on the Si-O-C ratio in OM particles, we estimated that 71%
of type 1-3 OM particles were associated with coal combustion. Our result suggests
that coal combustion in residential stoves was a widespread source from urban to rural
areas in the NCP. Average OM thickness which correlates with the age of the air
masses in type 6 particles only slightly increased from S3 to S2 to S1, suggesting that
the L&M hazes were usually dry (relative humidity < 60%) with weak
photochemistry and heterogeneous reactions between aerosols and gases. We
conclude that the direct emissions from these coal stoves without any pollution
controls in rural areas and in urban outskirts mainly contribute into the regional L&M
hazes in North China.





## 1 Introduction


Atmospheric particulate matter is composed of diverse chemical compounds,
both organic and inorganic matters. Organic aerosol particles are of two types:
primary organic aerosol (POA), directly emitted from fossil fuel combustion, biomass
burning, vehicular exhaust, and cooking; and secondary organic aerosol (SOA),
formed from the oxidation of gaseous volatile organic compounds (Kanakidou et al.,
2005). Organic aerosols account for 18-70% of the non-refractory submicron aerosol
particles in the atmosphere (Zhang et al., 2007). It is well known that organic aerosols
affect the atmosphere through the interaction with reactive trace gases, water vapor,
clouds, precipitation, and radiation (Fuzzi et al., 2006). Organic aerosols also
influence the physical and chemical properties of atmospheric particles (e.g., size,
light-absorptivity, and hygroscopicity); they directly affect visibility and climate by
scattering and absorbing solar radiation (Poschl, 2005;Kanakidou et al., 2005;Kulmala
et al., 2004). Although most organic aerosol components are known to have a cooling
effect on global climate, brown carbon in organic aerosols can absorb solar radiation
at shorter wavelengths and lead to warming (Alexander et al., 2008). Moreover, many
organic compounds (e.g., benzene, polycyclic aromatic hydrocarbons (PAHs),
toluene), which are toxic to human and other biological species have been found in
atmospheric particles (Mauderly and Chow, 2008).
In recent years, haze episodes have become one of the most serious environment
problems in China, following the rapid urbanization and population growth in eastern
China. China's Ministry of Environmental Protection on 1 January 2013, started to
monitor daily $PM_{2.5}$ concentrations and defined the various air pollution levels as:
excellent (0~35 μg m$^{-3}$), good (35~75 μg m$^{-3}$), light (75~115 μg m$^{-3}$), moderate
(115~150 μg m$^{-3}$), heavy (150~250 μg m$^{-3}$), and severe (> 250 μg m$^{-3}$) (HJ 663-2012).
Haze as a weather phenomenon is defined by visibility ≤ 10 km and RH ≤ 95%.
Previous studies have shown that haze levels normally are associated with $PM_{2.5}$
concentrations and RH (Shen et al., 2015;Wang et al., 2006;Chen et al., 2014). Based
on their results, we classify severe haze days (< 5 km) with $PM_{2.5}$ concentrations ≥



250 µg m$^{-3}$ and light (8-10 km) to moderate (5-8 km) haze days at 75-250 µg m$^{-3}$, both
with RH < 80%.
Severe wintertime haze pollution episodes were driven to a large extent by
secondary aerosol formation, and SOA were found to be of similar importance with
secondary inorganic aerosols (Guo et al., 2014;Huang et al., 2014). However, Sun et
al. (2013) showed that OM was dominant in wintertime haze pollution episodes but
POA concentrations were much higher than SOA in the aerosol particles. The reason
about the different results of OM particles among these studies could be the difference
of air pollution levels and RH during the haze pollution episodes. Severe winter hazes
were driven by stable synoptic meteorological conditions with high relative humidity
(i.e., RH > 80% frequently or consistently occurred during the haze episodes) (Tie et
al., 2015). Due to the dimming effect of high-concentration aerosol particles, an
enhanced production rate of secondary aerosols was suggested as an important
contribution from the complicated heterogeneous reactions (Li et al., 2011c;Zheng et
al., 2015). Because of severe haze episodes with unusually high PM$_{2.5}$ concentrations
(> 250 µg m$^{-3}$) and low visibility (< 5 km), many scientists have been investigated the
physicochemical properties of their aerosol particles (Huang et al., 2014;Guo et al.,
2014;Zheng et al., 2015). Although this knowledge is critical to understand severe
haze formation and its impacts on human health, the frequency of severe haze events
is low, they are of short duration. For example, we statistically analyzed haze days
during the winter (~ 92 days) of 2014-2015 in nine different cities. Figure S1 shows
that light and moderate (L&M) haze days occurred 22-63% of the time and that severe
haze days were less frequent at 4-32%, with the variation dependent on location
within the NCP. As we know, the L&M hazes are precursors of severe ones. Zheng et
al. (2015) suggested that characteristics of aerosol particles in severe hazes would not
be the same in L&M hazes. Compared to severe hazes, the L&M hazes were most
frequent in winter and lasted longer in the NCP. Therefore, understanding aerosol
particles in the more common L&M hazes in the NCP is important to further evaluate
their impacts on human health and regional climate.



107   Various ''bulk'' analytical instruments have been used to study organic aerosol

108  particles during haze episodes. High resolution time-of-flight aerosol mass

109  spectrometry (HR-AMS) was applied to determine the mass concentrations and bulk

110  composition of organic aerosols (Sun et al., 2010). Gas chromatography-mass

111  spectrometry (GC-MS) provided chemical composition and structures of organics in

112  aerosols (Fu et al., 2012;Wang et al., 2009). It should be noted that bulk analytical

113  techniques only provide average properties of $PM_{2.5}$ and the mixing state, phase, and

114  morphology of organic particles remain unknown. Detailed information about

115  individual organic particles, moreover, is critical to evaluate their formation, their

116  sources, and their hygroscopic and optical properties in the atmosphere. For example,

117  copious tar balls containing homogeneous BrC ocurr in the smoldering smoke from

118  biofuels (Chakrabarty et al., 2010;Adachi and Buseck, 2011;Alexander et al.,

119  2008;Chakrabarty et al., 2013;China et al., 2013;Hand et al., 2005;Posfai et al., 2004).

120  Atmospheric particles undergo liquid-liquid phase separations and go on to form OM

121  coatings on inorganic aerosol particles (You et al., 2012). The surface coating by OM

122  on individual particles influences water uptake and evaporation of inidividual

123  particles and their heterogeneous reactions in the atmosphere (Shiraiwa et al.,

124  2011;Zawadowicz et al., 2015;Riipinen et al., 2011). Despite the importance of these

125  phenomena, the morphology and mixing state of OM particles in wintertime L&M

126  hazes in the NCP have not been examined, although OM is dominant in fine particles

127  (Sun et al., 2013).

128   To characterize organic aerosols in greater detail in L&M hazes, individual

129  particles in the NCP in winter were analyzed using transmission electron microscopy

130  coupled with energy-dispersive X-ray spectroscopy (TEM/EDX), atomic force

131  microscopy (AFM), and Nanoscale secondary ion mass spectrometer (NanoSIMS).

132  Aerosol particles were simultaneously collected on TEM grids in the NCP during

133  13-23 December, 2014. Morphology, mixing state, and size of organic aerosols were

134  systematically characterized and compared at the three sampling sites (background

135  island site, mountain site, and urban site) in the same haze. This information enables



the discussion of source and ageing mechanisms of OM particles, which leads to
insights about the formation of regional wintertime L&M hazes in the NCP.
**2    Experimental Methods**
**2.1  Sampling sites and particle collection**
$PM_{2.5}$ and individual particle samples were simultaneously collected during
13-23 December, 2014 at three sampling sites: an urban site (S1), a mountain site (S2),
and a background site (S3) in the NCP (Fig. 1). S1- the urban site in Jinan (36.67 °N,
116.98 °E) is 50 km north of S2. S1 is a typical polluted city with high-density
residential areas surrounded by large industrial zones (Li et al., 2011c). Aerosol
particles collected at S1 mainly reflect local, ground-based urban and industrial
emissions. S2-Mt. Tai (1534 m a.s.l., 36.251 °N, 117.101 °E) is the highest mountain in
the middle of the NCP, ~230 km inland from the Bohai and Yellow Seas. S2 is the
perfect location to observe air pollutants near the planetary atmospheric layer over the
NCP. Aerosol particles collected at S2 represent regional transport (Li et al., 2011a).
S3-Changdao Island, the National Station for Background Atmospheric Monitoring
site (38.19 °N, 120.74 °E), is in the Bohai Sea. During the winter monsoon season, S3
is the downwind of the Jing-jin-ji area (i.e., Beijing city, Tianjin city, and Hebei
province) and Shandong province. Therefore, S3 serves as a polluted background site
from the transport of continental air. Therefore, aerosol particles collected at the three
sampling sites display the different pollutant characteristics of polluted urban air,
upper air layer, and background island air in the NCP.
$PM_{2.5}$ was collected on 90 mm quartz filters for 9-11.5 h using three KB-120
samplers at a flow rate of 100 L/min. The quartz filters were stored in a refrigerator
for OC, EC, and water soluble ion analysis. In the study, OC and EC concentrations of
70 quartz filters were analyzed by an OC/EC analyzer (Sunset Lab) and water-soluble
ions (i.e., $K^+$, $Na^+$, $Ca^{2+}$, $Mg^{2+}$, $NH_4^+$, $F^-$, $SO_4^{2-}$, $NO_3^-$, and $Cl^-$) by an ion
chromatography system (Dionex ICs-90). Three single-stage cascade impactors with a
0.5-mm diameter jet nozzle at a flow rate of 1.0 L/min were used to collect aerosols



onto copper TEM grids coated with carbon film (carbon type-B, 300-mesh copper,
Tianld Co., China). The collection efficiency of the impactor is 50% for particles with
an aerodynamic diameter of 0.25 μm and with a density of 2 g cm$^{-3}$ (Li et al., 2011a).
After sample collection, the Cu grids were placed in a sealed, dry and clean
environment until the TEM analysis. 11 aerosol samples at each sampling site were
selected and analyzed by the TEM.
**2.2 TEM analyses**
The JEOL JEM-2100 transmission electron microscopy operated at 200 kV with
energy-dispersive X-ray spectrometry (TEM/EDX) was used to analyze individual
particles. An energy-dispersive X-ray spectrometer (EDX) can detect elements
heavier than carbon. EDX spectra were executed for 15 s to minimize the potential
beam damage. TEM grids are made of copper (Cu), so the Cu element will be
excluded in the analyses. The distribution of particles on the TEM grids was not
uniform: coarser particles were deposited near the center and finer particles dispersed
on the fringe. To make sure that the analyzed particles were representative of the
entire size range, three to four areas were chosen from the center and periphery of the
sampling spot on each sample.
**2.3 NanoSIMS analysis**
Individual aerosol particles were analyzed using a nanoscale secondary ion mass
spectrometer (NanoSIMS) 50L, an ultrahigh vacuum technique for surface and
thin-film analysis at the Institute of Geology and Geophysics, Chinese Academy of
Sciences. In this study, $^{12}C^-$, $^{16}O^-$, $^{12}C^{14}N^-$, $^{14}N^{16}O_2^-$, $^{32}S^-$ ions in individual particles
were obtained when the Cs+ primary ion beam caused the ionization of atoms within
the particles. Furthermore, ion intensity mappings of individual particles with
nanometer resolution can show the distribution of different ions. $^{12}C^-$ and $^{12}C^{14}N^-$
represent the organic matter in individual particles (Chi et al., 2015;Ghosal et al.,
2014;Li et al., 2016a).
**2.4 AFM analysis**



Atomic force microscopy (AFM) with a tapping mode analyzed aerosol particles under ambient conditions. AFM, a digital NanoscopeIIIa Instrument, can determine the three dimensional morphology of particles. The AFM settings contain imaging forces between 1 and 1.5 nN, scanning rates between 0.5 and 0.8 Hz, and scanning range sizes at 10 μm with a resolution of 512 pixels per length. After the AFM analysis, composition of the same particles was confirmed by TEM, with 20, 25, and 13 individual particles analyzed by this method for each of the three sampling sites. The Nanoscope analysis software can automatically obtain bearing area (A) and bearing volume (V) of each analyzed particle according to the formulas described by Chi (Chi et al., 2015).

The relationship is shown in Figure S2 (EVD=0.8334ECD (S1), EVD=0.7286ECD (S2), and EVD=0.6601ECD (S3)). Therefore, the equivalent circle diameter (ECD, x) of individual aerosol particles measured from the iTEM software can be further converted into equivalent volume diameter (EVD, y) based on these relationships.

## 3. Results

### 3.1 Regional haze periods in North China Plain

Aerosol particles were collected in three regional L&M hazes during 13-23 December, 2014 (Fig. S3). Moderate Resolution Imaging Spectroradiometer (MODIS) images clearly display a regional haze layer covering the three sampling sites in the NCP (Fig. 1). The average $PM_{2.5}$ concentrations were 96.6 μg m$^{-3}$ (range: 79-171 μg m$^{-3}$), 88.6 μg m$^{-3}$ (range: 76-110 μg m$^{-3}$), and 80.3 μg m$^{-3}$ (range: 75-84 μg m$^{-3}$) on haze days, twice as high as on clear days (52, 48, and 32.3 μg m$^{-3}$) at S1, S2, and S3, respectively. The RH at all three sampling sites was lower than 60% during the sampling periods (Fig. S4).

Figure 1 shows similar back trajectories of air masses at S1, S2, and S3. Haze episodes started to form in the NCP when air masses changed from north to southwest or west. Therefore, the regional hazes in the NCP were controlled by the same





meteorological conditions (e.g., RH, temperature, and wind). The average
concentrations of OC, EC, OC/EC, water-soluble ions and their mass proportions in
$PM_{2.5}$ were much higher on haze days than on clear days at the three sampling sites
(Table S1). OC on haze days was more than 1.7 times higher than that on clear days at
three sites. OM concentration was estimated at 20-33 µg m$^{-3}$ and OM/$PM_{2.5}$ ratio was
at the range of 23-34% during haze days in the NCP (Table S1).
**3.2  Morphology of organic particles**
TEM/EDX identified five types of particles during the haze episodes: sulfates
(including K-rich sulfate and ammonium sulfate), fly ash, mineral, soot, and
C-dominated particles (Fig. S5). These results are consistent with previous studies
during the haze episodes in the NCP (Li et al., 2012;Li et al., 2011c). In order to
remove the interference of the carbon substrate on TEM grids, a nanoSIMS was
employed to verify C-dominated particles through $^{12}C^{14}N^{-}$ and $^{12}C^{-}$ mappings (Fig. 2
and Fig. S6). Figure 2 clearly shows that one near-spherical particle, which contains C,
O, and minor Si on TEM grids, displays strong $CN^{-}$ and $O^{-}$ signals but no clear $NO_2^{-}$
and $S^{-}$ signals. As a result, this type particle can be confirmed as the OM particle (Li
et al., 2016a;Ghosal et al., 2014). TEM analysis showed that OM-containing particles
were most abundant in all the haze samples, accounting for 70% of the 5090 analyzed
particles (Fig. S7).
Based on the morphology of OM particles, they were divided into six different
types: spherical (type 1, Fig. 3a), near-spherical (type 2, Fig. 3b)), irregular (type 3,
Fig. 3c), domelike (type 4, Fig. 3d), dispersed-OM (type 5, Fig. 3e), and OM-coating
(type 6, Fig. 3f). Because the high-resolution TEM images of individual particles can
clearly display particle interior mixing structures, it allows us to identify OM particles
based on their different shapes in OM-containing particles (Fig. S5). Figure 4 shows
that the proportions of type 1-3 in OM particles was 73%, following type 4 at 5%,
type 5 at 13%, and type 6 at 14%. Further, we measured the projected area, the
perimeter, the maximum projected length, and the maximum projected width of 967
selected OM particles. From these data, the sphericity (Sph) and aspect ratio (AR) of





different types of OM particles were calculated, which characterize their shape and
thereby imply their aging during transport and their emission sources (Li et al.,
2016b). The Sph and AR were defined by the following formulas referred to by (Li et
al., 2013).
*Aspect Ratio (AR)*. The maximum ratio of width and height of a bounding
rectangle for the measured object is the aspect ratio. An aspect ratio of 1 (the lowest
value) indicates a particle is not elongated in any direction.
*Sphericity (Sph)*. Sphericity describes the sphericity or "roundness" of the
measured object by using central moments. A sphericity of 1 (the highest value)
indicates a particle is perfectly spherical.
$$Sph = \frac{\sqrt{4\pi S}}{P} = \frac{\sqrt{4\pi\pi R_1^2}}{2\pi R_2} = \frac{R_1}{R_2} \tag{1}$$

$$AR = \frac{L_{max}}{W_{max}} \tag{2}$$

Where $S$ is projected area, $R_1$ is equivalent area radius, $P$ is perimeter, $R_2$ is
equivalent perimeter radius, $L_{max}$ is the maximum projected length, and $W_{max}$ is the
maximum projected width.
Table 1 displays the Sph and AR of individual OM particles measured by the
iTEM software. At the three different sampling sites, OM particles were in the fine
range with diameters $<$ 1 μm. The statistics show that spherical OM particles
exhibited the highest Sph at 0.96-0.99 and the lowest AR at 1.0-1.03 at the three sites,
followed by OM coating (Sph: 0.88-0.93, AR: 1.06-1.08), near-spherical OM (Sph:
0.82-0.83, AR: 1.12-1.13), domelike OM (Sph: 0.64-0.73, AR: 1.24-1.35), irregular
OM (Sph: 0.50-0.57, AR: 1.39-1.48) and dispersed-OM particles (Sph 0.49-0.58, AR:

272 1.35-1.46).

**3.3 Mixing state of OM particles**
Although we identified different types of OM particles in individual particles, 86%
were internally mixed with non-OM particles, such as soot, mineral, metal, fly ash,
and sulfate particles (Fig. S5). Based on their morphological mixing state, we





discriminated four OM internally mixed particles: OM-soot (Fig. 5a and Fig. S8),
OM-mineral/metal (Fig. 5b), OM-fly ash (Fig. 5c), and OM-sulfate particles (Fig.
5d-e). Our results show that 83% of type 1-4 OM particles were attached to soot,
mineral, sulfate, and metal particles, only 17% of type 1-4 OM particles were
externally mixed particles, and all the type 5-6 OM were internally mixed with sulfate
particles (Fig. S7).
Figure 6 shows number fractions of OM internally mixed particles in different
size bins from 0.04 to 4.5 μm at the three sampling sites. OM-soot particles
commonly occurred at S1 but they were mixed with certain amounts of sulfates at S2
and S3 during the sampling period (Fig. S8). The major OM internally mixed particles
include 45% OM-soot particles and 46% OM-sulfate particles at S1, 35% and 62% at
S2, 33% and 56% at S3 (Fig. S7). As a result, the number fraction of OM-sulfate
particles increases from S1 to S2 and S3. OM-soot containing particles dominated in
the finer size range (< 300 nm) at the three sampling sites (Fig. 6). In addition, 19% of
OM-sulfate particles were internally mixed with inclusions (i.e., fly ash and metal) at
all the three sampling sites (Fig. S7).
**3.4  Size distribution of OM-containing particles**
Figure 7 shows size distributions of OM-containing particles at the three
sampling sites. Aerosol particles collected at S2 and S3 display a similar peak at ~400
nm, much smaller than the peak at 600 nm at the S1 site (Fig. 7a). This result
indicates that sizes of locally emitted OM-containing particles are much larger than
the long-range transported OM-containing particles. We further obtain size
distributions of type 1-3 OM particles at the three sampling sites during one haze
episode. Interestingly, type 1-3 OM particles displayed similar peaks around 350 nm
at all three sampling sites (Fig. 7b). This result suggests that the OM sources were
similar under different transport ranges in the same haze layer over the NCP.

**4.  Discussion**
**4.1 Sources of OM-containing particles**





TEM adequately characterized the morphology and mixing state of
OM-containing particles in wintertime L&M hazes. We found that the type 1-3 OM
particles (Fig. 3a-c) were most abundant in the hazes and that most of them were
internally mixed with non-OM particles. This result is consistent with one previous
study which found abundant amorphous spherical OM particles in the outflow of a
haze plume in East Asia (Zhu et al., 2013). Moreover, Li et al. (2012) found large
amounts of type 1 OM particles in a coal-burning region in the China Loess Plateau in
winter. However, some studies found abundant type 5-6 OM particles in the
atmosphere. For example, Moffet et al. (2013) suggested that OM coating particles
become dominant following particle transport from an urban to background site and
they didn't report abundant type 1-3 OM particles in North America. Also, Adachi et
al. (2014) only reported that most sulfate particles were coated by secondary OM
coating in remote mountain air in Japan. Based on these comparisons, we conclude
that those type 1-3 OM particles were not emitted by vehicular emissions in the NCP.
It should be noted that recent studies did not find abundant type 1-3 OM particles
at three sampling sites on haze episodes in spring and summer (Li et al., 2011b;Yuan
et al., 2015). Zhang et al. (2008) suggested that industrial boilers had cleaner
combustion with much less by-product of particulate carbon and with much lower
levels of OM, while residential stoves had significantly higher emissions of
carbonaceous particulate matter with emission rates 100 times higher than that of
industrial boilers. As a result, we believe that the type 1-3 OM particles were not
emitted from heavy industries or coal-fired power plants but, instead, they were from
coal combustion or biomass burning for household heating and cooking in wintertime.
In particular, the abundant near-spherical OM particles with higher Sph and lower AR
indicate that these OM particles formed in cooling, polluted plumes from coal
combustion and biomass burning.
Biomass burning and coal combustion both can produce types 1-2 OM particles
and all contain a certain amount of Si beside C and O (Li et al., 2012;Posfai et al.,
2004;Hand et al., 2005;Adachi and Buseck, 2011). Li et al. (2012) found that primary





OM particles contain much higher Si from coal combustion than biomass burning. To
evaluate OM sources in this study, we compared ratios of Si, O, and C in individual
OM particles collected in haze and fresh OM particles from corn stalks and coal
combustion conducted in the laboratory (supplementary materials). Figure 8 clearly
shows that the Si ratio in individual OM particles is ordered as coal combustion >
haze particles > corn stalks; and that 71% of haze OM particles are associated with
coal combustion. Therefore, we can conclude that coal combustion contributes more
type 1-3 OM particles than biomass burning in the wintertime L&M haze. This result
is consistent with the source apportionment of OM particles based on their mass
concentrations (Elser et al., 2016;Sun et al., 2013). Furthermore, the consistent Sph
and AR of abundant type 1-3 OM particles at the three sampling sites (Table 1)
suggests that coal combustion in residential stoves was a widespread emission source
from urban to rural areas in the NCP. The vehicular emissions at S1 had a higher
contribution of fine soot (i.e., BC) particles with diameters < 300 nm (Fig. 6). Fly
ash/metal particles were normally considered from coal-fired power plant and heavy
industries (Li and Shao, 2009;Shi et al., 2003;S. X. Wang, 2010). Although fly
ash/metal-bearing particles occurred at three sampling sites but their fraction was only
19% (Fig. S7), suggesting that these large stationary sources were not major sources
for the primary particles.
**4.2 Ageing of OM particles**
The complicated mixing structures of individual particles can be used to evaluate
particle ageing mechanisms (Li et al., 2016b). Figure 1 suggests that S2 and S3 as the
polluted background sites received aged particles after long-range transport and that
the urban site of S1 received more fresh particles. Indeed, OM-soot particles at S2 and
S3 sites were internally mixed with sulfates but not at S1 (Fig. S8). To evaluate
particle ageing processes, we measured OM coating thickness in type 6 OM-coating
particles (e.g., Fig. 5e) because coating thickness on secondary particles can be used
to infer particle ageing during transport (Moffet et al., 2010;Moffet et al., 2013). In
this study, OM coating thickness increased with particle size and that their average





values at the three sampling sites were ordered as S3 > S2 > S1 (Fig. 9). The results
suggest that coarse secondary particles underwent more ageing than the fine particles
and that the particles at S3 underwent the most ageing. However, number fractions of
type 6 OM particles were small at three sampling sites (14.8% at S2 and 12% at S3,
and 2.9% at S1) (Fig. 9). The result indicates weak atmospheric reactions for SOA
formation in the whole haze layer.
**5.    Conclusions and atmospheric implications**

371        Abundant type 1-3 OM particles at S1, S2, and S3 suggested that coal

combustion and biomass burning used for cooking and heating in residential sector
in winter significantly contributed to the haze layer over the NCP. Although heavy
industrial and coal-fired plants emitted large amount of gases such as $SO_2$, $NO_x$, and
VOCs in the NCP (Wang et al., 2012;Zhang et al., 2015;Wang et al., 2013a), we
didn't observe many secondary organic and inorganic aerosols in wintertime L&M
hazes; these aerosols are common particle types in heavy haze and fog episodes (Li et
al., 2011c). The results indicated very weak photochemical reactions due to lower $O_3$
concentrations and weaker solar radiation, two constraints that reduce the conversion
of the acidic gases into aerosol particles (Ma et al., 2012). Also, the L&M hazes were
dry with RH < 60% (Fig. S4), which prohibits heterogeneous reactions between
aerosol and gases (Zheng et al., 2015). These reasons can explain why we found
higher number and mass fractions of abundant type 1-3 primary OM particles and
lower type 6 secondary OM particles in the hazes. This result is consistent with a
previous study using AMS (Sun et al., 2013), which showed 69% primary OM
particles and only 31% SOA in winter hazes. Concerning mixing processes, the sizes
(Fig. 7) and shapes (Sph and AR in Table 1) of the type 1-3 OM particles were
consistent with fresh particles at S1 and aged particles at S2 and S3. Therefore,
coagulation between OM and non-OM particles was expected to be the major mixing
mechanism on the dry haze with its weak photochemistry.

391        Once humidity exceeds 60% during a haze episode, secondary inorganic and

organic species from heterogeneous reactions can significantly contribute to haze



aerosols (Zheng et al., 2015). In a word, primary emissions from cooking and heating
likely caused the regional L&M haze episodes in the NCP. Moreover, coal
combustion and biomass burning can emit K-rich sulfates (e.g., Fig. 5d) (Wang et al.,
2013b). These residential coal stoves in rural areas and in the urban outskirts have no
pollution controls and directly emit particulate carbon and other pollutants. The
emission control of residential coarse coal combustion is simply not regulated by the
national environmental protection bureau, even though this bureau has made much
recent progress in controlling emissions from heavy industries and coal-fired power
plants.

402        We also noticed that a large proportion of type 1-3 primary OM particles has been

determined to be brown carbon (BrC), which can absorb shorter-wavelength solar
radiation (Alexander et al., 2008). The BrC and BC in the haze layers over the NCP
undoubtedly absorb solar energy and influence the vertical mixing of air. One
modeling study shows that these absorbing aerosols can lead to a more stable
atmospheric stratification over the NCP that decreases turbulence diffusion by 52%
and decreases the planetary boundary layer (PBL) height by 33 % (Wang et al., 2015).
OM particles from the direct emissions of coarse coal combustion and biomass
burning not only contributed to $PM_{2.5}$ in the L&M hazes, but also influenced haze
stability. Moreover, the type 1-3 OM particles from coal combustion in residential
stoves mainly consist of PAHs (Zhang et al., 2008). In this study, we found that type
1-3 OM particles not only occurred in 70% aerosol particles but also were
concentrated on fine particles (< 1 μm) (Fig. 7a). Therefore, these OM-containing
particles in the frequent L&M hazes could pose a threat to human health for one long
period throughout the winter. Although haze episodes commonly occur in the NCP,
the morphology and mixing state of OM particles in the wintertime L&M hazes are
largely different from those in the more severe wintertime hazes and from those in
L&M hazes in other seasons (Li and Shao, 2009;Li and Shao, 2010). Therefore, these
microscopic observations provide important information on the different haze
formations and evaluate their atmospheric impacts on climate and human health.




**Acknowledgments**

We appreciate Peter Hyde's comments and proofreading. This work was funded by
the This research was supported by National Key Project of MOST
(JFYS2016ZY01002213), National Natural Science Foundation of China (41575116),
Projects of International Cooperation and Exchanges, National Natural Science
Foundation of China (41571130033), Shandong Provincial Science Fund for
Distinguished Young Scholars, China (JQ201413), and Programs of Shandong
University (2014QY001/2015WLJH37).





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





# Figure Captions

**648**

**649** **Figure 1.** Cluster-means of backward trajectories at S1 (urban Jinan), S2 (Mt. Tai top),

**650** and S3 (Changdao island) sites during 13-23 December, 2014. Air masses on clear

**651** days normally are from the northwest and air masses on haze days are from the west

**652** or southwest. MODIS images show the layer during the light and moderate regional

**653** hazes over the NCP.

**654** **Figure 2.** NanoSIMS-based ion intensity mappings of $^{12}C^{14}N^-$, $^{14}N^{16}O_2^-$, $^{16}O^-$, and $^{32}S^-$

**655** from a near-spherical OM particle.

**656** **Figure 3.** Typical TEM images of different types of OM particles. (a) Type 1:

**657** spherical shape; (b) type 2: near-spherical shape; (c) type 3: irregular shape; (d) type 4:

**658** domelike OM (droplet-like particle); (e) type 5: dispersed-OM; (f) type 6:

**659** OM-coating

**660** **Figure 4.** Number fraction of OM types in particulate matter during light and

**661** moderate haze episodes from 13 to 23 December 2014. The 2562 OM particles were

**662** analyzed by TEM-EDX.

**663** **Figure 5.** Typical TEM images of OM internally mixed particles (a) a spherical OM

**664** particle attached to a soot particle; (b) a near-spherical OM particle attached to a

**665** mineral particle; (c) fly ash particles attached to a near-spherical OM particle; (d) OM

**666** mixed with sulfate (K)-fly ash particle and its sublimed particle under strong electron

**667** beam; (e) OM as a coating mixed with a sulfate (K, Na) particle. The element

**668** compositions in OM, mineral, fly ash, and sulfate particles were measured by the

**669** TEM/EDX.

**670** **Figure 6.** Number fraction of OM internally mixed particles at (a) S1 site, (b) S2 site,

**671** and (c) S3 site. The number of analyzed particles in different size ranges is shown

**672** above each column.

**673** **Figure 7.** Size distributions of OM-containing particles and OM particles during the

**674** same hazes. (a) Size distributions of 1088, 1341 and 1136 OM-containing particles

**675** collected at S1, S2, and S3. (b) Size distributions of 269, 173, and 297 type 1-3 OM

**676** particles collected at S1, S2, and S3.





**Figure 8.** Triangular diagram of weight ratios of Si-O-C based on TEM/EDX data. 34
OM particles from coal combustion and 28 OM particles from corn stalks combustion,
and 281 OM particles produced from the haze samples in this study. The three lines
represent the links of the ratios of Si/O and Si/C in haze, corn stalks and coal
combustion samples, respectively.
**Figure 9.** The relationship between the size of individual particles and their sulfate
cores based on 366 OM-coating particles at S1, S3, and S3 sites. The smaller slope
represents the thicker OM coating. The number fractions of OM coating to
OM-containing particles at three sampling sites are shown in the pie charts.

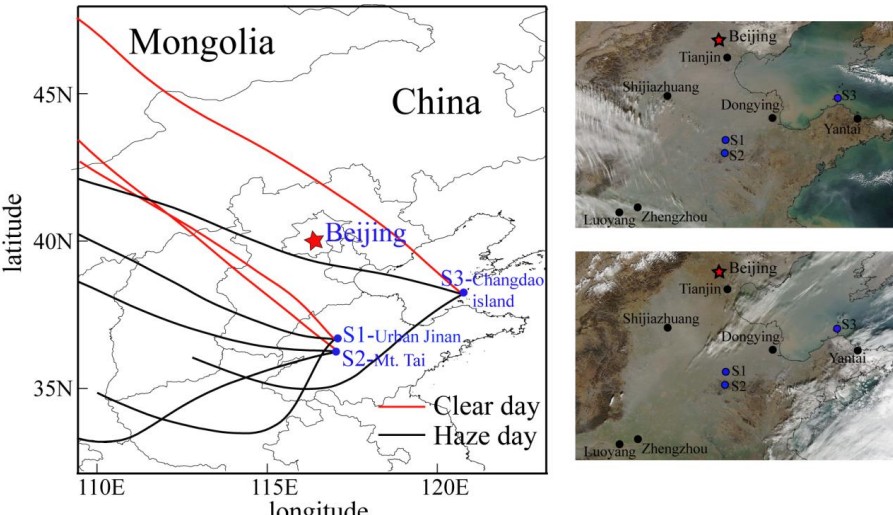

**Figure 1.** Cluster-means of backward trajectories at S1 (urban Jinan), S2 (Mt. Tai top), and S3 (Changdao island) sites during 13-23 December, 2014. Air masses on clear days normally are from the northwest and air masses on haze days are from the west or southwest. MODIS images show the layer during the light and moderate regional hazes over the NCP.

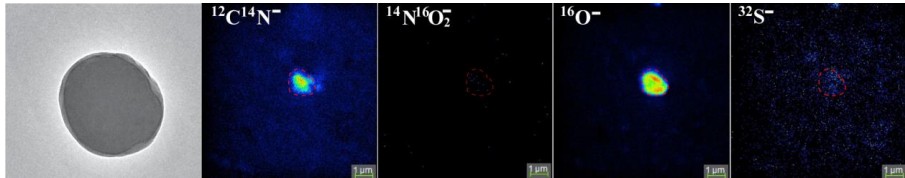

**Figure 2.** NanoSIMS-based ion intensity mappings of $^{12}C^{14}N^-$, $^{14}N^{16}O_2^-$, $^{16}O^-$, and $^{32}S^-$ from a near-spherical OM particle.



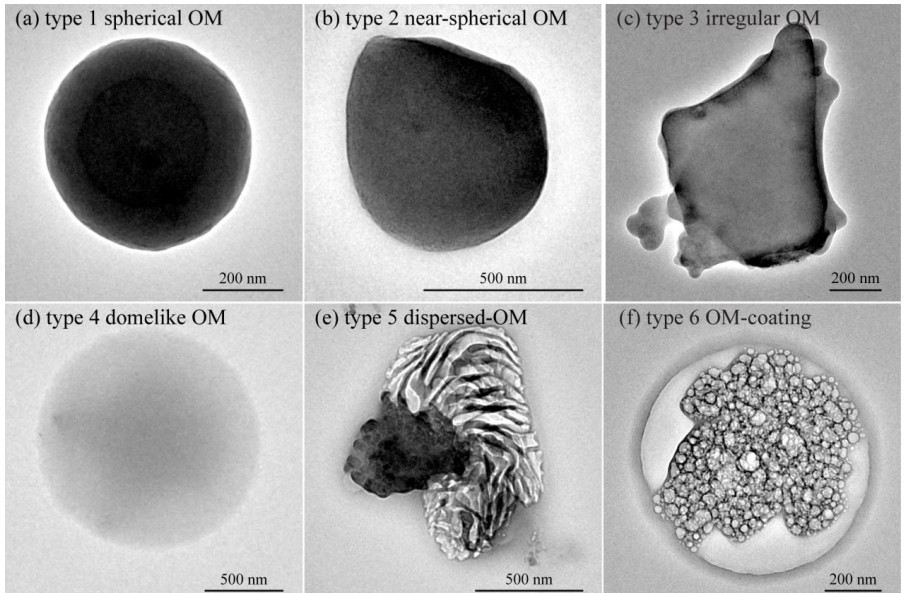

**Figure 3.** Typical TEM images of different types of OM particles. (a) Type 1: spherical shape; (b) type 2: near-spherical shape; (c) type 3: irregular shape; (d) type 4: domelike OM (droplet-like particle); (e) type 5: dispersed-OM; (f) type 6: OM-coating

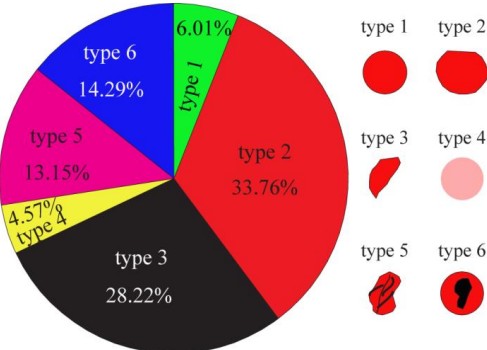

**Figure 4.** Number fraction of OM types in particulate matter during light and moderate haze episodes from 13 to 23 December 2014. The 2562 OM particles were analyzed by TEM-EDX.





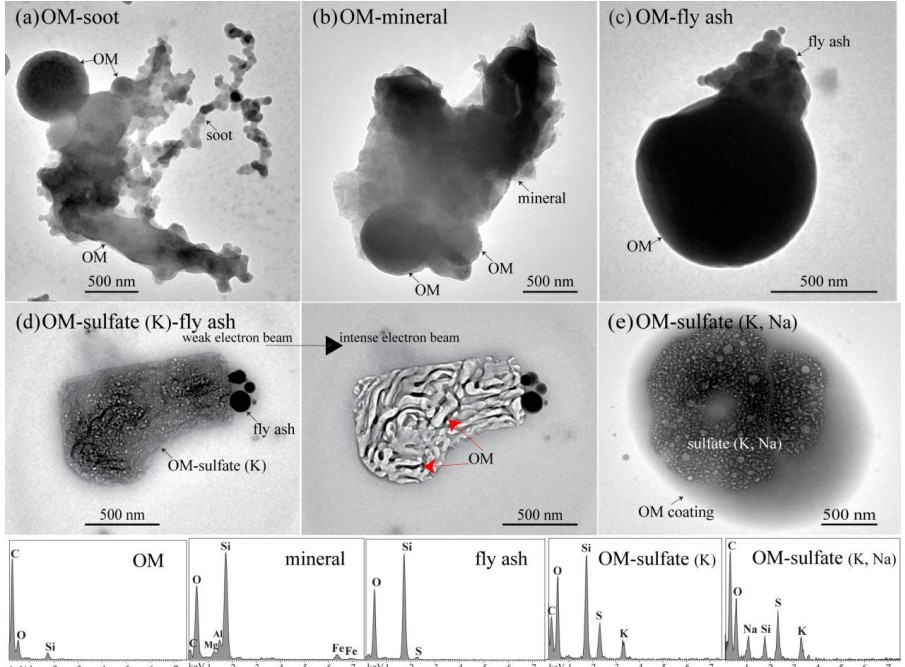

**Figure 5.** Typical TEM images of OM internally mixed particles (a) a spherical OM particle attached to a soot particle; (b) a near-spherical OM particle attached to a mineral particle; (c) fly ash particles attached to a near-spherical OM particle; (d) OM mixed with sulfate (K)-fly ash particle and its sublimed particle under strong electron beam; (e) OM as a coating mixed with a sulfate (K, Na) particle. The element compositions in OM, mineral, fly ash, and sulfate particles were measured by the TEM/EDX.

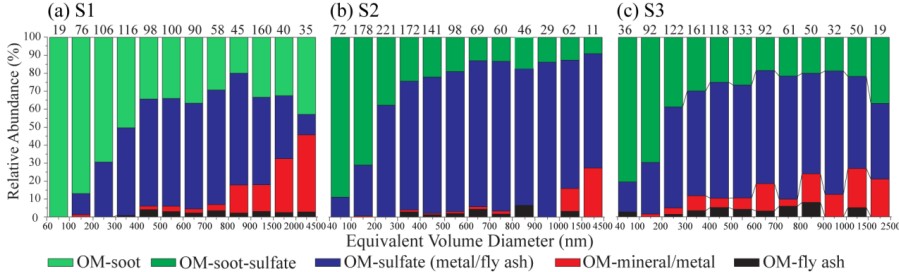




**Figure 6.** Number fraction of OM internally mixed particles at (a) S1 site, (b) S2 site, and (c) S3 site. The number of analyzed particles in different size ranges is shown above each column.

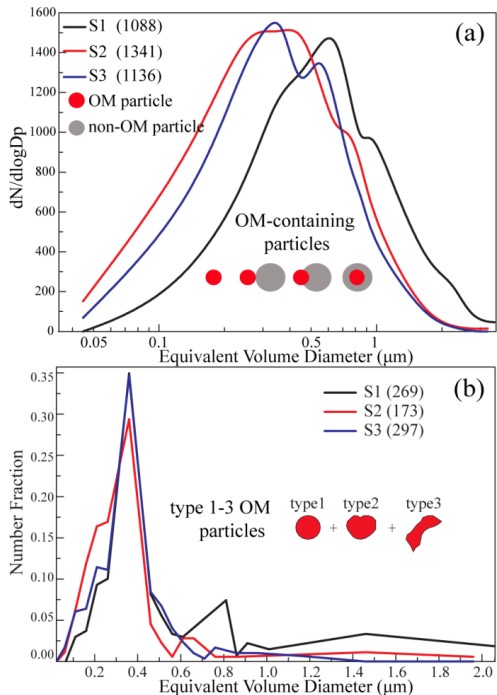

**Figure 7.** Size distributions of OM-containing particles and OM particles during the same hazes. (a) Size distributions of 1088, 1341 and 1136 OM-containing particles collected at S1, S2, and S3. (b) Size distributions of 269, 173, and 297 type 1-3 OM particles collected at S1, S2, and S3.





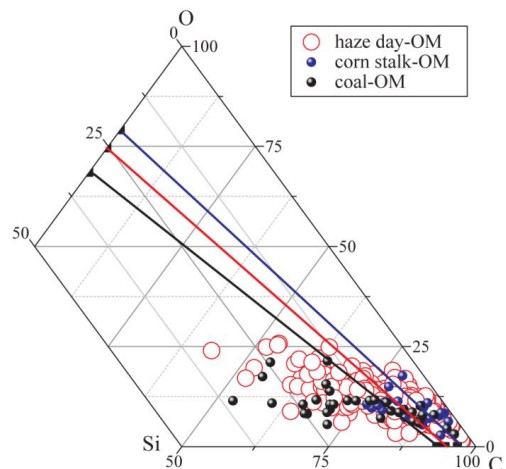

**Figure 8.** Triangular diagram of weight ratios of Si-O-C based on TEM/EDX data. 34 OM particles from coal combustion and 28 OM particles from corn stalks combustion, and 281 OM particles produced from the haze samples in this study. The three lines represent the links of the ratios of Si/O and Si/C in haze, corn stalks and coal combustion samples, respectively.

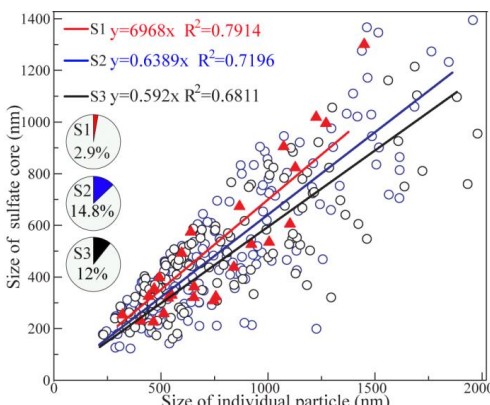

**Figure 9.** The relationship between the size of individual particles and their sulfate cores based on 366 OM-coating particles at S1, S3, and S3 sites. The smaller slope represents the thicker OM coating. The number fractions of OM coating to OM-containing particles at three sampling sites are shown in the pie charts.



**Table 1.** Average size, number, sphericity, and aspect ratio for different OM types at the three sampling sites.

| Sampling site | Type | Average Size(nm) | Number | Average Sph (min, max); (±stdev.) | Average AR (min, max); (±stdev.) |
|---|---|---|---|---|---|
| S1 | Type1 spherical | 407.8 | 18 | 0.9664 (0.8, 1.0); (±0.05) | 1.0223 (1.0, 1.16); (±0.03) |
| | Type2 near spherical | 348.5 | 79 | 0.8172 (0.50, 1); (±0.11) | 1.1275 (1.0, 1.41); (±0.08) |
| | Type3 irregular | 538.68 | 151 | 0.5046 (0.07,0.95); (±0.21) | 1.4764 (1.04, 3.57); (±0.41) |
| | Type4 domelike | 758.76 | 23 | 0.6374 (0.29, 0.9); (±0.21) | 1.3958 (1.08, 1.89); (±0.26) |
| | Type5 dispersed-OM | 750.8 | 64 | 0.4940 (0.1, 0.92); (±0.21) | 1.4646 (1.06, 2.85); (±0.40) |
| | Type6 OM-coating | 672.18 | 12 | 0.9251(0.45, 1); (±0.13) | 1.0627 (1, 1.45); (±0.11) |
| S2 | Type1 spherical | 282.15 | 22 | 0.9628 (0.76, 1.0); (±0.06) | 1.0281 (1.00, 1.25); (±0.05) |
| | Type2 near spherical | 323.15 | 68 | 0.8254 (0.53, 1.0); (±0.12) | 1.1197 (1.00, 1.37); (±0.09) |
| | Type3 irregular | 399.26 | 62 | 0.5746 ((0.06,0.96); (±0.24) | 1.4102 (1.04, 2.79); (±0.42) |
| | Type4-domelike | 511.35 | 25 | 0.7341 (0.15, 1.0); (±0.29) | 1.2433 (1.00, 2.28); (±0.36) |
| | Type5-dispersed-OM | 846.99 | 34 | 0.5773 (0.1, 0.9); (±0.25) | 1.4218 (1.08, 2.71) (±0.45) |
| | Type6-OM-coating | 775.53 | 66 | 0.8791 (0.26, 1) (±0.22) | 1.0797 (1, 1.57) (±0.12) |
| S3 | Type1 spherical | 391.5 | 27 | 0.9901 (0.92, 1.0); (±0.02) | 1.0065 (1.0, 1.06); (±0.02) |
| | Type2 near spherical | 373.09 | 122 | 0.8341 (0.33, 1.0); (±0.13) | 1.12 (1,1.51); (±0.10) |
| | Type3 irregular | 374.85 | 117 | 0.5675 (0.09, 0.96); (±0.21) | 1.392 (1.04, 2.82); (±0.34) |
| | Type4 domelike | 705.94 | 14 | 0.6325 (0.22, 1); (±0.23) | 1.3519 (1, 1.857); (±0.22) |
| | Type5 dispersed-OM | 574.62 | 35 | 0.5495 (0.4,0.65); (±0.19) | 1.3475 (1.21, 1.51); (±0.31) |
| | Type6 OM-coating | 828.35 | 28 | 0.9291 (0.36,1); (±0.30) | 1.0591 (1.0, 1.60); (±0.14) |