# Peer review of "Direct observations of organic aerosols in common wintertime hazes in North China: insights into direct emissions from Chinese residential stoves"

_Atmospheric Chemistry and Physics, 2016_

## Referee Comment (RC1) · Anonymous Referee #1 · 11 Aug 2016

This paper has presented a descriptive scheme of individual organic aerosol particles for the light and moderate hazes which were often seen in Northern China Plain. The mixing states and size distribution of the organic aerosol particles were discussed, and it is proposed that most of the organic aerosol particles were sourced from the uncontrolled domestic coal combustion. A number of the-state-of-the-art techniques were used in the characterization of individual particles, including TEM, nanoSIMS, and AFM. The results obtained are interesting and helpful for us to understand the forming mechanisms of the wintertime light and moderated hazes. I agree this paper to be accepted for publication after moderate revision.

My detailed comments on current manuscript are as follows:

[Figure]

1. Selection of sampling sites needs to be introduced in more details, especially the importance and significance of the three sites. I understand that the Jinan city is regarded as the representative of uncontrolled coal combustion site, but in fact the urban area of this megacity might not rely on coal for domestic energy anymore and instead, the petrol and gas might be its major energy sources. This manuscript is designed to explain domestic coal combustion in the vast area of countryside, but no this type of sampling sites were selected.

2. "1 Introduction" part: the introduction needs to be succinct and should be more concentrated on the organic aerosol particles and related hazes. The current text in the introduction part is too complicated and not well focused on the main aims.

3. The classification of the six types of organic particles needs to be careful. For example, the type 4 particle (doom-like) looks more like a mixture of possible organic and other materials such as ammonia or nitrates, and the type 5 particle (dispersed) may be the results of the shrinkage of organic-coated particle.

4. Line 30: "(Tai, S2)" might be "(Mt. Tai, S2),"

5. Line 33: I suggest to change "OM-coating" into "coating OM" for the type 6 particle.

6. Line 71: "the various air pollution levels" may be changed into "the various air quality levels"

7. Line 74: the definition of "Haze as a weather phenomenon is defined by visibility $\leq$ 10 km and RH $\leq$ 95%" requires references.

8. Line 117 and also throughout the whole manuscript, acronyms and abbreviations must be explained at first occurrence. For example, the first appeared "BrC" should have a full word phrase.

9. Line 129-133: the methods mentioned here are repetitive of the "2 Experimental Methods" part.

10. "2.1 Sampling sites and particle collection": In Line 145, the authors mentioned that "aerosol particles collected at S1 mainly reflect local, ground-based urban and industrial emissions". This means the S1 site can't represent the potential uncontrolled coal combustion source?

11. Line 151: "During the winter monsoon season, S3 is the downwind of the Jing-Jin-Ji area ….and Shandong province." This looks not correct. From the map, S3 is located in the east of the JJJ area, and how can we regard it is the downwind of S1 and S2? Furthermore, S3 is not located upwind area, it may not appropriate to serve as a background (clean) site.

12. "2.3 NanoSIMS analysis" part: It is good to see that the NanoSIMS gives the ions 12C- and 12C14N- which could represent the organic matter in individual particles. However, to the study of this manuscript, how many of the individual particles were analyzed? Were all particles measured by NanoSIMS?

13. Line 198: "with 20, 25, and 13 individual particles analyzed by this method for each of the three sampling sites." Which sites exactly these numbers correspond to?

14. Line 229-230: Soot may also be the C-dominated particles?

15. Line 254: Please check if it is "…ratio of width and height.." or ""…ratio of height and width…".

16. 275-276: The category of "soot, mineral, metal, fly ash, and sulfate particles" is not same as that of the line 229-230?

17. Line 278: The OM-fly ash might be the overlapped particles during sampling and not necessarily the mixed particle in the air?

18. Line 314, "For example, Moffet et al. (2013) suggested ……Based on these comparisons, we conclude that those type 1-3 OM particles were not emitted by vehicular emissions in the NCP". However, all these data for comparisons were from North America and Japan, which I don't think can exclude the type 1-3 OM particles present from

vehicular emissions in urban areas of Chinese Cities.

19. Line 325, the authors didn't analyze the emissions from heavy industries or coal-fired power plants, so I don't think they can obtain the conclusion that "the type 1-3 OM particles were not emitted from heavy industries or coal-fired power plants" and that "they were from coal combustion or biomass burning for household heating and cooking in wintertime". More evidence needs to be provided.

20. Line 413: Please check if the 1-3 OM occupy 70% of aerosol particles or 70% of the organic particles?

21. "5. Conclusions and atmospheric implications" needs to be simplified, and what are major conclusions?

22. "Acknowledgments": There are some repetitive words between line 424 and line 425.

23. Table 1: The decimal number should keep consistent.

24. English of the text needs to be polished by a native English speaker.
* * *

---

## Referee Comment (RC2) · Anonymous Referee #2 · 9 Sep 2016

The manuscript presented the physicochemical characterizations of aerosol particles collected during haze events in China from single particle analysis using TEM, AFM, and nanoSIMS. Based on the characteristics of particles analyzed by TEM, the manuscript discussed the classification of particles and their size, morphology, and mixing state with complement analysis from AFM and nanoSIMS. The manuscript then discussed the transport and aging of particles and their possible sources. The subjects of this manuscript are within the scope of ACP. The following comments should be addressed before it can be considered to publish in this journal.

General comments:

1, Although the observations of individual particles probed by TEM can provide certain information to relate to the sources, the conclusions regarding the exact sources should be carefully evaluated. It could be expected that the sources during such haze events are complicate. To exclude the other sources and limit it to residential stove emission, additional information is needed to support such conclusion. For example, as discussed in the manuscript, there are two particle types, OM-fly ash and OM-sulfate(K)-fly ash as shown in Figure 5, what are the possible sources of these fly ash containing particles, what are the composition of these fly ash? Is it possible that these are from coal-fired plants or industrial emissions? In Line 142, as indicated by the authors that S1 can be influenced by the industrial emissions. The statements regarding the sources in Abstract and Conclusion section should be reworded if no further supporting evidences is provided to constrain the sources.

2, The Introduction and Conclusion sections should be revised. In the Introduction section, there is a long discussion between the severe haze and L&M haze events and their differences, but later there is no comparison or further discussion in the main text. This part should be shortened. It may be beneficial to reader or to the context to focus more on single particle analysis or source characterization. The Conclusion and Atmospheric Implication section can be more concise and focus more on the findings from these observations. For example, the discussion of BrC and the implications, for such discussion should be limited to a certain extend unless data are provided showing the OM particles are BrC.

3, The classification of the particle types is not very clear and straightforward in the current form. It is suggested to add a figure or table to describe how the particles are grouped. Some definitions are not consistent. For example, the "OM-coating (type 6)" in L33, "C-dominated" in L230, "OM-containing particles" in L237, "OM coating" in L269, these terms are confusing and not consistent throughout the text. In Figure 3 (e), the type 5 dispersed-OM is very similar to the particle in Figure5(d) which was bleached by the beam, why it is classified as OM particle which seems to have S and K? In Figure 3, for the type 6, what is the chemical composition of cores? Should type

6 belong to OM internally mixed particles as showing in Figure5?

Specific comments:

1, L31, use "morphologies"?

2, L37, what does "cooling, polluted plumes" mean?

3, L38, what kind of detector is used for EDX analysis? Please justify the use of "Si-O-C ratio" to estimate to contribution of coal combustion.

4, L44, "aerosols" means "particles and gases"

5, L46, use "to" instead of "into"?

6, L58, the sentence should be revised.

7, L61 , "Poschl" should be "PÖschl"

8, L70-71, It is suggested to use "Ministry of Environmental Protection of People's Republic of China" . I guess the Ministry of environmental protection is not monitoring the PM2.5 by itself. Please revise the statement.

9, L73, please use the right document citation, what is "HJ 663-2012"?

10, L75-76, use "associated with different levels of PM2.5 concentrations and RH"?

11, L93, delete "been"

12, L94 delete "their"

13, L98 delete "different"

14, section 2.1, more details should be provided regarding the sampling procedure and sample sites. What was the sample height at each site, what about the sampling time and duration?

15, L155, use "represent" instead of "display"

16, L163, the impactors are used to collect particles not aerosols

17, L168, one sample for each day was analyzed? What is the sampling duration for each grid?

18, L174, use"acquired"?

19, L183, what is brand and model for NanoSIMS?

20, L218-221, it is not clear what the authors try to discuss. Please revise.

21, L253, please revise how the reference is cited.

22, L301-302, The sentence is not clear.

23, L318, there is no sufficient evidence to support this conclusion.

24, L326-331, this section should be carefully revised as discussed in the General Comments.

25, L333, what is the EDX detector; detector background may contribute to Si signal?

26, L339, as it is shown in Figure 8, the ratios are not following the lines, the data points are sort of deviating from the lines. Please discuss in more details.

27, L363-365, This is not very clear. Considering the scatting of the data points, is it significantly different among these three cases? It would be more straightforward if coating thickness is calculated and compared among these sites with statistical test.

28, L368, the statement is not convincing if only consider the number fractions of type 6 OM particles.

29, L393, "in other words"?

Figure 1, are the two black lines for each site indicating the range of the backward trajectories?

Figure 6, it is not easy to distinguish the green colors for the OM-soot and OM-sootsulfate.

Figure 9, "OM coating", do you mean "OM-coating particles"?
* * *

---

## Referee Comment (RC3) · Anonymous Referee #3 · 12 Sep 2016

Review of "Direct observations of organic aerosols in common wintertime hazes in North China: insights into their size, shape, mixing state, and source" by Chen et al. This is a well-written paper that presents single particle analyses of samples obtained during moderate haze events in North China during winter of 2014 at an urban, island, and mountain site. The analyses focused on the composition, size, morphology, and mixing state of organic particles (OM). Single particle analysis is critical to understanding the properties of OM in the atmosphere and for assessing modeling efforts to estimate their contribution to visibility degradation and climate change. This paper furthers our knowledge of OM morphology and is a useful contribution to the literature regarding OM properties, especially during moderate hazes. I recommend publication

after major revisions that address the concerns and comments listed below.

A main concern is the use of carbon data obtained from EDX on TEM grids to comment on source emissions (coal, haze, corn).The contribution of carbon and oxygen from the TEM grids makes the C and O EDX data semiquantitative at best, yet the authors are using them to determine source profiles and to derive conclusions without a mention of the interference other than line 232. Also, the main motivation for the paper is the contribution of these particles to haze, yet no mention of optical properties of these particles was included. Can the authors extend their results/discussion to the impact of their results on our knowledge of OM optical properties? They have data on particle shape, for example, and could comment on the assumptions used to model optical properties of OM particles (typically assumed to be spherical). In addition, comparisons to bulk data would be helpful to understand whether OC/EC ratios and inorganic/OM fractions are reflected in the particle type/morphology. Finally, several arguments and conclusions provided in the paper would benefit from clarification and additional evidence; details are in the comments below.

Comments Line 22: Qualify this statement with "North China" after "haze episodes" because this statement is generally not true- many studies have focused on many different levels of hazes.

Line 22 : "freqent" is a typo.

Line 72: Define "PM2.5" at first usage.

Line 80: Add "in China" after "episodes"

Line 99: How is a haze day defined with respect to time? How long do high concentrations or poor visibility have to last to be considered an episode? How different are the timescales for moderate versus heavy haze days?

Line 117: Define "BrC" at first usage.

Line 122: "inidividual" is a typo.

Line 131: Were the same TEM grids used for all three analyses?

Line 142: Please provide elevations of S1 and S3.

Line 152: Remove "the" from between "is" and "downwind"

Line 157: Please provide more detail regarding the choice of 9-11.5 hour sampling period. Was this sampling repeated continuously? Or was it repeated daily only at the same time each day?

Line 164: Did the TEM grid sampling occur on the same sampling schedule as the bulk sampling?

Line 168: How were the 11 aerosol samples chosen? What time periods did the samples correspond to?

Line 170: What was the order of the analysis for the three methods? How was destruction to particles from electron beams or vacuum minimized in the order of the analysis?

Line 202: Define EVD and ECD at first use.

Line 211: What time periods to the MODIS images correspond to?

Line 213-214: It is not clear what time periods averages correspond to? All periods above 75 $\mu$g/m3?

Line 226: Point out that although the concentrations increased between haze and clear days, the fraction of PM2.5 that is organic did not change that much. It appeared that the fraction of organics and inorganics remained fairly stable regardless of higher haze events.

Line 232:Was nanoSIMS performed on all TEM grids so that the carbon content of the particles could be confirmed this way? Obtaining carbon contribution from TEM grids using TEM/EDX is obviously very uncertain given the interference from the grids.

Line 237: The interference of the grid makes determining OM content of particles from TEM qualitative at best. How is this avoided with this analysis?

Line 240: Were the OM particle morphology characterized subjectively? Meaning, did a single user determine the type of each particle based on visual inspection, or was this somehow determined by a computer algorithm?

Line 242: Can the authors provide some additional description of the "domelike" particles? What does this "domelike" structure imply?

Line 243: Did any of these OM-type particles behave differently under the beam or vacuum?

Line 245: For which site?

Line 261: It would be clearer if the equation for AR was moved up into the "Aspect Ratio" section.

Line 286: This information would fit better in the previous paragraph.

Line 283-292: According to line 166, the D50 for this sampler is 0.25 $\mu$m. Was a collection efficiency applied to the data to account for this? If not the size information should be considered qualitative at best (especially since the maximum bin is 4.5 $\mu$m). Some mention of this should be made in this section. Are the bin widths greater than the uncertainty in the size data? To assist with the interpretation of Figure 6, the data from all sites should have the same size bins and scale on the figures.

Line 294: Parts (a) and (b) would be more easily compared if they had the same scale. Part (a) has a log-scale and (b) does not, so the size distributions are difficult to compare. Also, what is the significance of the OM-containing particle diagrams within part (a)? Does part (a) include all OM-containing particle types (1-6) while part (b) only include the subset 1-3? Which haze event does figure 7 correspond to? What does the bimodal peak around 0.8 um correspond to in part (b)?

[Figure]

Line 314: Change "coating" to "coated"

Line 316: Where in North America?

Line 318-9: I am not sure how this conclusion follows from the previous comparisons?

Line 327-328: State how OM 1-3 particles from coal-combustion from power plants and residential heating/cooking would differ that leads to this conclusion. What additional evidence?

Line 336: I have several questions/concerns from the data presentation and analysis in this paragraph. First, including and comparing C and O from the TEM/EDX analysis here is concerning given the interference from the grids. I am not sure that carbon data are very meaningful in this context. I see the description of how Si-O-C line for haze determined from the supplemental, but some mention should also be included in the paper. Haze can correspond to very different particle composition and would not likely have a single Si-O-C ratio. Is corn combustion representative of biomass burning in the region during this time of year? Again, I am not convinced these data are meaningful given the C and O interference.

Line 345: The sphericity of OM 1-3 particles does not necessarily suggest that these emissions are from coal combustion. Many other studies have reported on spherical OM particles that originated from biomass burning. Shape alone does not necessarily correspond to emission type.

Line 347: I suggest restating that the vehicular emissions at S1 led to higher contributions of soot particles because no mention of vehicular emissions at S1 has been made up to now. Instead, one might infer that the high contribution of soot particles at S1 could likely be from vehicular emissions in an urban area. Figure 6 shows fly-ash as part of two different types (OM-sulfate metal/fly ash and OM-fly ash) and how is that reconciled with the contributions shown in Figures S7? If the presence of fly ash is the evidence used for large stationary sources, than this designation should be made

earlier (see comment for line 327).

Line 356: The back trajectories for all sites look similar during haze events, so I am not convinced that aging can be determined separately for sites based on Figure 1.

Line 365: Define "coarse"

Line 376: How would a secondary organic particle appear in the TEM analysis? One might argue that the mixed OM-sulfate or the coated particles are secondary in nature. I also disagree with the statement that not many inorganic aerosols were observed given that Table S1 states the inorganic fraction of PM2.5 was actually higher than TC/PM2.5 at all sites.

Line 381: Many hygroscopicity studies have demonstrated water associated with particles at RH values less than 60%.

Line 386: This statement seems inconsistent with line 377 that states that SOA are common in heavy haze but only 31% in winter hazes. What type of hazes?

Line 394: Recall from Figure 1 that back trajectories suggest different transport on haze days. Are the authors trying to state that cooking and heating only from other regions are influencing the hazes? Can the authors reconcile and clarify this argument?

Line 402: Some comment here on the bulk OC and EC data and comparisons to the single particle results would be useful. Is the relative abundance of soot particles on haze days consistent with higher EC measurements? Is this also true for OM-containing particles?

Line 410: The influence of direct emissions on haze stability has not been established in this paper.

Line 413: What does "70% aerosol particles" mean?

Line 417: Transport must be taken into account when making the statements regarding differences in moderate and heavy hazes, as well as meteorological controls such

as boundary layer depth, wind speed, etc. Heavier hazes could be associated with stagnant conditions when pollution builds up, but emissions could be the same.

Figures/Tables: Figure 1: What dates do the MODIS images correspond to?

Figure 6: As mentioned in the comments, the figure would be more easily compared if the bin widths and figure scales were the same.

Figure 7. Provide the significance of the OM-containing particle diagram within part (a); it can be interpreted a few different ways. Convert the x-scale in part (b) to log and use the same scale as part (a), same as with the y-axis.

Figure 9: Does size here refer to EVD? How was sulfate core size measured? Table 1: It is not necessary to report so many digits for the average sphericity and AR values; only significant digits are necessary.

---

## Author Comment (AC1) · 5 Oct 2016

**Direct observations of organic aerosols in common wintertime hazes in North China: insights into direct emissions from Chinese residential stoves**

Chen et al.,

We appreciated the reviewers' comments which significantly improve quality of the manuscript. We carefully answer them one by one as below.

1. Selection of sampling sites needs to be introduced in more details, especially the importance and significance of the three sites. I understand that the Jinan city is regarded as the representative of uncontrolled coal combustion site, but in fact the urban area of this megacity might not rely on coal for domestic energy anymore and instead, the petrol and gas might be its major energy sources. This manuscript is designed to explain domestic coal combustion in the vast area of countryside, but no this type of sampling sites were selected.

   Answer: We added more information about sampling sites. We admit that air pollutants in Jinan city represented complicated sources such as industrial, vehicular, and residential emissions. If the reviewer noticed the finding of the whole paper, you can find aerosol particles collected in weak photochemical environment in winter. In other words, most of aerosols were primary particles instead of less secondary particles transformed from $SO_2$, NOx, and VOCs. Although the industrial and vehicular emission could be dominant, they mostly emitted trace gases (e.g., $SO_2$, NOx, and VOCs). Among these major sources in Jinan city, coal combustion in residential stoves only emitted large amounts of primary particles, in particular, they emitted abundant primary organic particles.

2. "1 Introduction" part: the introduction needs to be succinct and should be more concentrated on the organic aerosol particles and related hazes. The current text in the introduction part is too complicated and not well focused on the main aims.

   Answer: We revised introduction. Please notice that we deleted large part in second paragraph.

3. The classification of the six types of organic particles needs to be careful. For example, the type 4 particle (domelike) looks more like a mixture of possible organic and other materials such as ammonia or nitrates, and the type 5 particle (dispersed) may be the results of the shrinkage of organic-coated particle.

   Answer: In this study, we used EDX to examine chemical composition of individual particles. If the type 4 particle contain ammonia or nitrates (except organic nitrate), these particles should not be stable under the electron beam. We found that these particles were stable in the TEM. The type 5 particle is other important particle types which is different from organic-coated particles. The particle types have been well described in our latest paper in JGR-Atmos. which is under review.

   About the type 4 particle, we collect similar particles from coal combustion in stoves in laboratory. Based on our observations in previous studies and laboratory experiments, we can surely that the classification is no problem.

4. Line 30: "(Tai, S2)" might be "(Mt. Tai, S2),"

   Answer: We revised this to Mt. Tai

5. Line 33: I suggest to change "OM-coating" into "coating OM" for the type 6 particle.

   Answer: To make consistent with our previous study, we want keep the OM-coating particles. We used the particle name in one new paper under review in JGR.

6. Line 71: "the various air pollution levels" may be changed into "the various air quality levels"

   Answer: We changed the "pollution" to "quality".

7. Line 74: the definition of "Haze as a weather phenomenon is defined by visibility $\leq$ 10 km and RH $\leq$ 95%" requires references.

   Answer: We revised this sentence and added reference for this definition. Please see Line 75.

8. Line 117 and also throughout the whole manuscript, acronyms and abbreviations must be explained at first occurrence. For example, the first appeared "BrC" should have a full word phrase.

   Answer: We added the full name of BrC: "brown carbon (BrC)".

9. Line 129-133: the methods mentioned here are repetitive of the "2 Experimental Methods" part.

   Answer: We deleted the repetitive sentence and revised this part. Please see Line 120.

10. "2.1 Sampling sites and particle collection": In Line 145, the authors mentioned that "aerosol particles collected at S1 mainly reflect local, ground-based urban and industrial emissions". This means the S1 site can't represent the potential uncontrolled coal combustion source?

    Answer: Please see our first reply (1).

11. Line 151: "During the winter monsoon season, S3 is the downwind of the Jing-Jin-Ji area … and Shandong province." This looks not correct. From the map, S3 is located in the east of the JJJ area, and how can we regard it is the downwind of S1 and S2? Furthermore, S3 is not located upwind area, it may not appropriate to serve as a background (clean) site.

    Answer: Maybe the reviewer misunderstood our meaning. We change words to make more clearly here. We didn't address that S3 site is downwind S1 and S2. Here, three different sampling sites represent different environmental functions. In this study, S3 is treated as a polluted background site instead of the normal background site.

12. "2.3 NanoSIMS analysis" part: It is good to see that the NanoSIMS gives the ions $12C^-$ and $12C14N^-$ which could represent the organic matter in individual particles. However, to the study of this manuscript, how many of the individual particles were analyzed? Were all particles measured by NanoSIMS?

    Answer: The NanoSIMS is very complex and expensive instrument. It is not necessary to examine all the organic particles. The NanoSIMS only confirmed the organic findings from TEM. The routine procedures for this study will be shown as below.

    Firstly, we used the TEM to observe organic particles.

    Secondly, we selected some typical samples which contain large amounts of organic

particles for NanoSIMS experiments.

13. Line 198: "with 20, 25, and 13 individual particles analyzed by this method for each of the three sampling sites." Which sites exactly these numbers correspond to?
    Answer: We added sites here. Please see Line 199.

14. Line 229-230: Soot may also be the C-dominated particles?
    Answer: We agreed and changed "C-dominated" to "OM-like" particles.

15. Line 254: Please check if it is "…ratio of width and height…" or ""…ratio of height and width…".
    Answer: We revised this sentence to "ratio of length and width…" in Line 267.

16. Line 275-276: The category of "soot, mineral, metal, fly ash, and sulfate particles" is not same as that of the line 229-230?
    Answer: We corrected them and added the metal in Line 230.

17. Line 278: The OM-fly ash might be the overlapped particles during sampling and not necessarily the mixed particle in the air?
    Answer: The sampling duration was controlled to avoid overlap among different particles on the substrate. If the reviewer carefully looks at Figure 5c, you will find the connection between OM and fly ash which could not form during sampling process.

18. Line 314, "For example, Moffet et al. (2013) suggested : : :: : :Based on these comparisons, we conclude that those type 1-3 OM particles were not emitted by vehicular emissions in the NCP". However, all these data for comparisons were from North America and Japan, which I don't think can exclude the type 1-3 OM particles present from vehicular emissions in urban areas of Chinese Cities.
    Answer: Thank you very much. We added more explanation here including our previous studies in urban air and remote mountain air. Please see Line 351.

19. Line 325, the authors didn't analyze the emissions from heavy industries or coal fired power plants, so I don't think they can obtain the conclusion that "the type 1-3 OM particles were not emitted from heavy industries or coal-fired power plants" and that "they were from coal combustion or biomass burning for household heating and cooking in wintertime". More evidence needs to be provided.
    Answer: Same to the above reply. We added more explanation here.

20. Line 413: Please check if the 1-3 OM occupy 70% of aerosol particles or 70% of the organic particles?
    Answer: We revised this to "70% OM-containing aerosol particles". Please see Line 453.

21. "5. Conclusions and atmospheric implications" needs to be simplified, and what are major conclusions?

Answer: We revised and shortened the "conclusions and atmospheric implications" section.

22. "Acknowledgments": There are some repetitive words between line 424 and line 425.
    Answer: We removed the extra words here.

23. Table 1: The decimal number should keep consistent.
    Answer: We unified the data format in Tables.

24. English of the text needs to be polished by a native English speaker.
    Answer: We invited a native English speaker (Dr. Peter Hyde) to polish the English writing.

---

## Author Comment (AC2) · 5 Oct 2016

**Direct observations of organic aerosols in common wintertime hazes in North China: insights into direct emissions from Chinese residential stoves**

Chen et al.,

We appreciated the reviewers' comments which significantly improve quality of the manuscript. We carefully answer them one by one as below.

1. Although the observations of individual particles probed by TEM can provide certain information to relate to the sources, the conclusions regarding the exact sources should be carefully evaluated. It could be expected that the sources during such haze events are complicate. To exclude the other sources and limit it to residential stove emission, additional information is needed to support such conclusion. For example, as discussed in the manuscript, there are two particle types, OM-fly ash and OM sulfate(K)-fly ash as shown in Figure 5, what are the possible sources of these fly ash containing particles, what are the composition of these fly ash? Is it possible that these are from coal-fired plants or industrial emissions? In Line 142, as indicated by the authors that S1 can be influenced by the industrial emissions. The statements regarding the sources in Abstract and Conclusion section should be reworded if no further supporting evidences is provided to constrain the sources.

   Answer: We really appreciated your comments. Spherical fly ash typically contains Si and O which indicates coal-fired power plant. In the revised manuscript, we largely revised section 4.1 (Sources of OM-containing particles) and added more references. We also slightly revised abstract and conclusion.

2. The Introduction and Conclusion sections should be revised. In the Introduction section, there is a long discussion between the severe haze and L&M haze events and their differences, but later there is no comparison or further discussion in the main text. This part should be shortened. It may be beneficial to reader or to the context to focus more on single particle analysis or source characterization. The Conclusion and Atmospheric Implication section can be more concise and focus more on the findings from these observations. For example, the discussion of BrC and the implications, for such discussion should be limited to a certain extend unless data are provided showing the OM particles are BrC.

   Answer: We revised and shortened the discussion between the severe haze and L&M haze events. We deleted most parts suggested by the reviewer.

3. The classification of the particle types is not very clear and straightforward in the current form. It is suggested to add a figure or table to describe how the particles are grouped.

   Answer: The classification of particle types were based on their chemical composition and morphology in the TEM/EDX, and then calculated their shapes in the computer algorithm. We added Figure 2 to describe the flow chart of particle classifications.

4. Some definitions are not consistent. For example, the "OM-coating (type 6)" in L33, "C-dominated" in L230, "OM-containing particles" in L237, "OM coating" in L269, these terms are confusing and not consistent throughout the text.

   Answer: We revised the name of "C-dominated" to "OM-like". TEM could not exactly determine OM particles, so called OM-like particles before we showed the NanoSIMS result in the context.

Because OM particles were internally mixed particles, we used OM-containing particles to represent all the OM-related particles. OM-coating represents one OM mixing structure in the internally mixed particles. The individual aerosol particles classifications were shown in Figure 2.

5.  In Figure 3 (e), the type 5 dispersed-OM is very similar to the particle in Figure5 (d) which was bleached by the beam, why it is classified as OM particle which seems to have S and K?
    Answer: We revised Figure 2 to show the classification here. Figure 5d shows the OM-mixed containing S and K belong to the classifying rules in this study.

6.  In Figure 3, for the type 6, what is the chemical composition of cores?
    Answer: The cores in type 6 are secondary inorganic components such as sulfate and nitrate. Please see the Figure 5f, the core is sulfate.

7.  Should type 6 belong to OM internally mixed particles as showing in Figure 5?
    Answer: Yes, It belongs to OM internally mixed particles as shown in Figure 5. We revised the Figure 2 to show how we classified the particles.

8.  L31, use "morphologies"?
    Answer: We revised this word.

9.  L37, what does "cooling, polluted plumes" mean?
    Answer: We revised this sentence to: "formed in cooling process after polluted plumes emitted from…". Please see Line 38.

10. L38, what kind of detector is used for EDX analysis? Please justify the use of "Si-O-C ratio" to estimate to contribution of coal combustion.
    Answer: The EDX is from oxford instruments. TEM/EDX only can obtain semi-quantitive data for elements. Therefore, we could not make any significant conclusion from the each element in individual particles. However, it is significant to make comparisons of Si-O-C in many OM particles detected under the same TEM/EDX, which can avoid some impacts from the substrate or instrument. The same method has been used in Li et al., JGR, 2012 and Posfai et al., JGR, 2004 in the reference list.

11. L44, "aerosols" means "particles and gases"
    Answer: We revised "aerosols" to "particles".

12. L46, use "to" instead of "into"?
    Answer: We revised this word.

13. L58, the sentence should be revised.
    Answer: We revised this sentence. Please see Line 59.

14. L61 , "Poschl" should be "PÖschl"
    Answer: We revised this name.

15. L70-71, It is suggested to use "Ministry of Environmental Protection of People's Republic of China". I guess the Ministry of environmental protection is not monitoring the PM2.5 by itself. Please revise the statement.

Answer: We revised this to "Ministry of Environmental Protection of People's Republic of China".

16. L73, please use the right document citation, what is "HJ 663-2012"?

Answer: We revised this citation.

17. L75-76, use "associated with different levels of PM2.5 concentrations and RH"?

Answer: We revised this sentence to "haze levels normally are associated with different levels of PM2.5 concentrations and RH" in Line 77.

18. L93, delete "been"

Answer: We revised this sentence to "physicochemical properties…have been well understood". Please see Line 83.

19. L94 delete "their"

Answer: Please see the answer for comment 18. We revised this sentence.

20. L98 delete "different"

Answer: We deleted this "different" in line 90.

21. section 2.1, more details should be provided regarding the sampling procedure and sample sites. What was the sample height at each site, what about the sampling time and duration?

Answer: Thanks, we provide it in supplement as Table S2.

22. L155, use "represent" instead of "display"

Answer: We revised "display" to "represent ".

23. L163, the impactors are used to collect particles not aerosols

Answer: We changed "aerosols" to "particles" here.

24. L168, one sample for each day was analyzed? What is the sampling duration for each grid?

Answer: Yes, We added the information about analyzed samples in Table S2.

25. L174, use "acquired"?

Answer: We revised the word to "acquired".

26. L183, what is brand and model for NanoSIMS?

Answer: We added the brand and model "NanoSIMS 50L, CAMECA Instruments, Geneviers, France". Please see Line 185.

27. L218-221, it is not clear what the authors try to discuss. Please revise.

Answer: Here, we deleted descriptions about back trajectories in this manuscript.

28. L253, please revise how the reference is cited.

Answer: We revised it in line 262.

29. L301-302, The sentence is not clear.

Answer: We revised this sentence like this: "This result suggests that the type 1-3 OM sources were similar in the same haze layer over the NCP". Please see Line 333.

30. L318, there is no sufficient evidence to support this conclusion.

Answer: We added more explanation here including our previous studies in urban air and remote mountain air. Please see Line 351.

31. L326-331, this section should be carefully revised as discussed in the General Comments.

Answer: We revised this section and added other references for this conclusion.

32. L333, what is the EDX detector; detector background may contribute to Si signal?

Answer: No, the background from EDX detector has been calibrated to remove the possible influence. As we used EDX to detect the background film, we could not detect the Si signal. It should not be worried about that.

33. L339, as it is shown in Figure 8, the ratios are not following the lines, the data points are sort of deviating from the lines. Please discuss in more details.

Answer: Thank you for your comments. As you know, we obtained the data from internally mixed particles. Elements from OM in the EDX data should not be like pure OM or influenced by other aerosol components mixed in individual particles. Therefore, it should have one range like in Figure 8.

34. L363-365, This is not very clear. Considering the scatting of the data points, is it significantly different among these three cases? It would be more straightforward if coating thickness is calculated and compared among these sites with statistical test.

Answer: As our result, it is not significantly different among these three sites. We added average coating thickness in Figure 9.

35. L368, the statement is not convincing if only consider the number fractions of type 6 OM particles.

Answer: Yes, we revised this sentence. Please see Line 414.

36. L393, "in other words"?

Answer: We deleted this sentence here.

37. Figure 1, are the two black lines for each site indicating the range of the backward trajectories?

Answer: As other reviewer requests, we deleted backward trajectories in the Figure 1.

38. Figure 6, it is not easy to distinguish the green colors for the OM-soot and OM-soot sulfate.

Answer: We revised the color in Figure 6.

39. Figure 9, "OM coating", do you mean "OM-coating particles"?

Answer: Yes, We revised this. Please see the description for this Figure in Line 419.

---

## Author Comment (AC3) · 5 Oct 2016

**Direct observations of organic aerosols in common wintertime hazes in North China: insights into direct emissions from Chinese residential stoves**

Chen et al.,

We appreciated the reviewers' comments which significantly improve quality of the manuscript. We carefully answer them one by one as below.

1. Comments Line 22: Qualify this statement with "North China" after "haze episodes" because this statement is generally not true- many studies have focused on many different levels of hazes.
   Answer: We added "North China" after "haze episodes".

2. Line 22 : "freqent" is a typo.
   Answer: We revised this word to "frequent".

3. Line 72: Define "PM$_{2.5}$" at first usage.
   Answer: We added the definition of PM$_{2.5}$ (aerodynamic equivalent diameter $\leq$ 2.5 μm). Please see Line 71.

4. Line 80: Add "in China" after "episodes"
   Answer: We added them.

5. Line 99: How is a haze day defined with respect to time? How long do high concentrations or poor visibility have to last to be considered an episode? How different are the timescales for moderate versus heavy haze days?
   Answer: We added one statistic data for an example to show occurrence of haze episodes and the timescales (Figure S2).
   We statistically analyzed frequency of haze episodes in winter for nine cities. If one heavy haze episode persists more three days, the government will have the highest alert (red). Figure S2 shows that only two severe haze days occurred in wintertime. However, the light and moderate haze episodes are common and persistent longer (Figure S2).

6. Line 117: Define "BrC" at first usage.
   Answer: We added the full name of BrC. Please see Line 108.

7. Line 122: "inidividual" is a typo
   Answer: We revised this word.

8. Line 131: Were the same TEM grids used for all three analyses?
   Answer: 33 TEM grids were analyzed for TEM/EDX analysis. Three typical samples (one grid for each site) were chosen for AFM and NanoSIMS analysis because of the consistency of samples.

9. Line 142: Please provide elevations of S1 and S3.
   Answer: We added the elevations of S1 and S3.

10. Line 152: Remove "the" from between "is" and "downwind"

Answer: Deleted

11. Line 157: Please provide more detail regarding the choice of 9-11.5 hour sampling period. Was this sampling repeated continuously? Or was it repeated daily only at the same time each day?

   Answer: We added the details about sampling periods (daytime: 7:30-19:00 and nighttime: 19:30-7:00 (next day)) and revised this sentence. Please see Line 156.

12. Line 164: Did the TEM grid sampling occur on the same sampling schedule as the bulk sampling?

   Answer: Yes. It should be noticed that different samples have different sampling duration. Individual particle samples must be collected in a short time. We added more information to explain it. Please see Table S2 which includes the details of samples.

13. Line 168: How were the 11 aerosol samples chosen? What time periods did the samples correspond to?

   Answer: We added the time periods and other information for samples at three sites in Table S2. The selected samples as much as possible represent the whole hazes.

14. Line 170: What was the order of the analysis for the three methods? How was destruction to particles from electron beams or vacuum minimized in the order of the analysis?

   Answer: The order of analysis is TEM, AFM, and NanoSIMS. Some particles (e.g., sulfate and nitrate) can be destroyed under the electron beams in TEM, but particles in other areas of the same sample still keep well. AFM doesn't destroy the samples. Finally, we used NanoSIMS to analyze the same samples. Because the TEM grids must install on the special plate in NanoSIMS, we cannot take them back anymore. We used the special TEM grids with letters which can help us to find locations. The method is the best way to integrate three different analyzed instruments for the same samples.

15. Line 202: Define EVD and ECD at first use.

   Answer: We added the definitions of EVD and ECD and added their formulas in supplementary material.

16. Line 211: What time periods to the MODIS images correspond to?

   Answer: We added the date. These two MODIS images were got on December 14 and 19, respectively.

17. Line 213-214: It is not clear what time periods averages correspond to? All periods above 75 $\mu$ g/m3?

   Answer: Yes, all the haze periods were above $75 \mu g/m^3$ here. Please see Figure S4.

18. Line 226: Point out that although the concentrations increased between haze and clear days, the fraction of $PM_{2.5}$ that is organic did not change that much. It appeared that the fraction of organics and inorganics remained fairly stable regardless of higher haze events.

   Answer: We revised this section and we added that "the fraction of OC to $PM_{2.5}$ remained fairly stable regardless of L&M haze and clear days". Please see Line 223.

19. Line 232: Was nanoSIMS performed on all TEM grids so that the carbon content of the particles could be confirmed this way? Obtaining carbon contribution from TEM grids using TEM/EDX is obviously very uncertain given the interference from the grids.

Answer: No, we could not do all the TEM grids. We just chose typical OM particles to confirm their chemical ions. We admitted TEM/EDX obtained uncertain carbon contribution, but it doesn't influence our classification based on all the elemental compositions of individual particles. The method is quite normal for individual aerosol analysis in TEM and SEM (e.g., Li et al., JGR, 2012; Moffet et al., ACP, 2010)

20. Line 237: The interference of the grid makes determining OM content of particles from TEM qualitative at best. How is this avoided with this analysis?

Answer: We used morphology and EDX data both to identify OM particles, and then we can account their number faction.

21. Line 240: Were the OM particle morphology characterized subjectively? Meaning, did a single user determine the type of each particle based on visual inspection, or was this somehow determined by a computer algorithm?

Answer: We made such a classification firstly based on visual inspection and then made their shape by a computer algorithm. I think the potential user can to identify the OM particles as this way.

22. Line 242: Can the authors provide some additional description of the "domelike" particles? What does this "domelike" structure imply?

Answer: Here we only define it based on their morphology. We suspect the domelike particles are organic gels. As we have known that organic gel is a type of material which is translucent. Indeed, we found similar particles from biomass burning and coal used in residential stoves in the laboratory experiments.

23. Line 243: Did any of these OM-type particles behave differently under the beam or vacuum?

Answer: In the TEM, these analyzed OM particles behaved stable. Obviously, they were non-volatile OM.

24. Line 245: For which site?

Answer: We made the three sampling sites together for analysis.

25. Line 261: It would be clearer if the equation for AR was moved up into the "Aspect Ratio" section.

Answer: We moved up the equation into the AR section.

26. Line 286: This information would fit better in the previous paragraph.

Answer: Yes, we moved it to previous paragraph and revised this sentence. Please see Line 301.

27. Line 283-292: According to line 166, the D50 for this sampler is 0.25 μm. Was a collection

efficiency applied to the data to account for this? If not the size information should be considered qualitative at best (especially since the maximum bin is 4.5 µ m). Some mention of this should be made in this section. Are the bin widths greater than the uncertainty in the size data? To assist with the interpretation of Figure 6, the data from all sites should have the same size bins and scale on the figures.

Answer: We didn't consider the sampling efficiency. We know that the sampler should have higher loss efficiency, so we used size bins to make possible comparisons. Otherwise, the particle number cannot be direct compared. We revised the size bins in figure 6.

28. Line 294: Parts (a) and (b) would be more easily compared if they had the same scale. Part (a) has a log-scale and (b) does not, so the size distributions are difficult to compare. Also, what is the significance of the OM-containing particle diagrams within part (a)? Does part (a) include all OM-containing particle types (1-6) while part (b) only include the subset 1-3? Which haze event does figure 7 correspond to? What does the bimodal peak around 0.8 um correspond to in part (b)?

Answer: Figure 7a represents the size distribution of individual particles in all L&M haze episodes during sampling period in NCP. Please noticed we covert the particle number N into dN/dlogDp in the y-axis, so the x-axis can be considered as log mode. Figure 7b only represents size of type 1-3 OM in one haze event. The y-axis is real number faction so x-axis should use the normal size. Figure 7b correspond to one haze event on December 14-15 (we added it in the Figure caption).

We checked all the data and found the peak is too low to give more strong information for particle sources.

29. Line 314: Change "coating" to "coated"
Answer: We deleted this sentence.

30. Line 316: Where in North America?
Answer: We deleted this description here.

31. Line 318-9: I am not sure how this conclusion follows from the previous comparisons?
Answer: We added some references (Li and Shao, 2010) and (Li et al., 2015) in this part. They found only a few type 1-3 OM particles in urban and remote mountain air in China. Based on the comparison, we conclude that the type 1-3 OM particles were not directly emitted by vehicular emissions in the NCP.

32. Line 327-328: State how OM 1-3 particles from coal-combustion from power plants and residential heating/cooking would differ that leads to this conclusion. What additional evidence?
Answer: In our previous studies, we studied aerosol particles associated with power plants, they didn't emit spherical OM. We revised the section and added other references for this conclusion.

33. Line 336: I have several questions/concerns from the data presentation and analysis in this paragraph. First, including and comparing C and O from the TEM/EDX analysis here is concerning given the interference from the grids. I am not sure that carbon data are very

meaningful in this context. I see the description of how Si-O-C line for haze determined from the supplemental, but some mention should also be included in the paper. Haze can correspond to very different particle composition and would not likely have a single Si-O-C ratio. Is corn combustion representative of biomass burning in the region during this time of year? Again, I am not convinced these data are meaningful given the C and O interference.

Answer: TEM/EDX only can obtain semi-quantitive data for elements. Therefore, we could not make any significant conclusion from the each element in individual particles. However, it is significant to make comparisons of Si-O-C in many OM particles detected under the same TEM/EDX, which can avoid some impacts from the substrate or instrument. The same method has been used in Li et al., JGR, 2012 and Posfai et al., JGR, 2004 in the reference list.

We added some description in the paper about the Si-O-C line for haze.

Thank you for your comments. As you know, we obtained the data from internally mixed particles. Elements from OM in the EDX data should not be like pure OM or influenced by other aerosol components mixed in individual particles. Therefore, the data points could not perfectly along with the lines. It should have one range like in Figure 8.

In the NCP, farmers harvest their corn in autumn and storage these corn stalks to burn in wintertime.

34. Line 345: The sphericity of OM 1-3 particles does not necessarily suggest that these emissions are from coal combustion. Many other studies have reported on spherical OM particles that originated from biomass burning. Shape alone does not necessarily correspond to emission type.

Answer: We agreed your comments. We revised the discussion about type 1-3 OM sources in the revised manuscript and deleted the description here between their shapes and sources. Please see Line 370.

35. Line 347: I suggest restating that the vehicular emissions at S1 led to higher contributions of soot particles because no mention of vehicular emissions at S1 has been made up to now. Instead, one might infer that the high contribution of soot particles at S1 could likely be from vehicular emissions in an urban area.

Answer: We deleted the descriptions about the sources of soot and fly ash particles here and focus on the OM particles in the revised manuscript.

36. Figure 6 shows fly-ash as part of two different types (OM-sulfate metal/fly ash and OM-fly ash) and how is that reconciled with the contributions shown in Figures S7?

Answer: Fly ash is a tracer of coal-fired power plant and heavy industrial, so it is an important kind of particle. Figure 2 shows only 19% of OM-sulfate particles were mixed with fly ash/metal. In Figure 2, we made OM-fly ash particles into a class and OM-sulfate mixed with fly ash into OM-sulfate group.

37. If the presence of fly ash is the evidence used for large stationary sources, than this designation should be made earlier (see comment for line 327).

Answer: Yes, we deleted the discussion about sources of fly ash and soot particles, but focus on the main sources of OM particles.

38. Line 356: The back trajectories for all sites look similar during haze events, so I am not convinced that aging can be determined separately for sites based on Figure 1.

Answer: We deleted the back trajectories because they only represent the air masses above 1500 m. We added more description about possible sources or location of the sampling sites.

39. Line 365: Define "coarse"

Answer: We revised this sentence. Please see Line 410-411.

40. Line 376: How would a secondary organic particle appear in the TEM analysis? One might argue that the mixed OM-sulfate or the coated particles are secondary in nature. I also disagree with the statement that not many inorganic aerosols were observed given that Table S1 states the inorganic fraction of $PM_{2.5}$ was actually higher than $TC/PM_{2.5}$ at all sites.

Answer: Thank you for your comments. Please notice that many OM-mixed particles more or less contain secondary inorganic species, which can be reflected in the classified names. The OM-coating particles cannot represent all the OM-sulfate particles. From the TEM observation, the OM-coating particle is much less in the samples collected in L&M hazes in winter than our previous study in summer or severe hazes in winter.

41. Line 381: Many hygroscopicity studies have demonstrated water associated with particles at RH values less than 60%.

Answer: Maybe it is true in some locations. We used the RH value for haze in North China from the reference (Zheng et al., 2015).

42. Line 386: This statement seems inconsistent with line 377 that states that SOA are common in heavy haze but only 31% in winter hazes. What type of hazes?

Answer: They did two researches in different haze levels and we added the haze type. Please see Line 438.

43. Line 394: Recall from Figure 1 that back trajectories suggest different transport on haze days. Are the authors trying to state that cooking and heating only from other regions are influencing the hazes? Can the authors reconcile and clarify this argument?

Answer: Your suggestion is very good, and we revised the Figure 1.

We did back trajectories at 1500 m for all sites in Figure 1 which could not represent ground pollutants' transportations.

44. Line 402: Some comment here on the bulk OC and EC data and comparisons to the single particle results would be useful. Is the relative abundance of soot particles on haze days consistent with higher EC measurements? Is this also true for OM-containing particles?

Answer: As the other reviewer's comments, we deleted the paragraph. The comparison can be done but we didn't focus on the EC in this study. Also, we want to mention that the particle classification cannot reflect the all the aerosol components. In the aged particles, many soot particles were internally mixed with OM and sulfate.

45. Line 410: The influence of direct emissions on haze stability has not been established in this

paper.

Answer: Thanks, we deleted this part.

46. Line 413: What does "70% aerosol particles" mean?

Answer: We revised this sentence to "OM-containing aerosol particles". Please see Line 453.

47. Line 417: Transport must be taken into account when making the statements regarding differences in moderate and heavy hazes, as well as meteorological controls such as boundary layer depth, wind speed, etc. Heavier hazes could be associated with stagnant conditions when pollution builds up, but emissions could be the same.

Answer: We agreed with your comments. The different haze levels must associate with meteorological data. Here we only focused on OM particle morphology.

48. Figure 1: What dates do the MODIS images correspond to?

Answer: We added the specific date of the MODIS images.

49. Figure 6: As mentioned in the comments, the figure would be more easily compared if the bin widths and figure scales were the same.

Answer: We revised this Figure's size bins.

50. Figure 7. Provide the significance of the OM-containing particle diagram within part (a); it can be interpreted a few different ways. Convert the x-scale in part (b) to log and use the same scale as part (a), same as with the y-axis.

Answer: Thank you for your advice. These two figures have different purpose. Figure 7a shows size distribution of the analyzed particle number. Please notice we also convert the y-axis using dN/dLogDp, so the log mode can make clear size distribution in the limited particle number. In contrast, Figure 7b only show number fraction with particle size. If we use log mode, the peaks will not clear anymore. Therefore, we do not change the x-scale in Figure 8b

51. Figure 9: Does size here refer to EVD? How was sulfate core size measured?

Answer: In Figure 9, particle size refers to EVD. We directly measure sulfate core size using the iTEM software (details in sections 2.2 and 2.4), and convert into EVD.

52. Table 1: It is not necessary to report so many digits for the average sphericity and AR values; only significant digits are necessary.

Answer: We revised Table 1.

---

## Author Comment (AC4) · 5 Oct 2016

**We added Figure 2 and modified other Figures as below:**

[Figure]

**Figure 1.** Regional haze layer covering the North China Plain: S1 (urban Jinan), S2 (Mt. Tai top), and S3 (Changdao island) sites. MODIS images on December 14 and 19 show grey haze layer during the light and moderate regional hazes over the NCP.

[Figure]

**Figure 2.** Flow chart of individual aerosol particles classification in L&M haze episodes in NCP based on TEM/EDX. 5090 individual particles were analyzed using TEM/EDX.

[Figure]

**Figure 6.** Number fractions of OM internally mixed particles at (a) S1 site (urban Jinan), (b) S2 site (Mt. Tai top), and (c) S3 site (polluted background). The number of analyzed particles in different size ranges is shown above each column.

[Figure]

**Figure 9.** The relationship between the size of individual particles and their sulfate cores based on 366 OM-coating particles at S1, S3, and S3 sites. The smaller slope represents the thicker OM coating. The number fractions of OM coating particles to OM-containing particles at three sampling sites are shown in the pie charts.

**Supplementary Figures:**

[Figure]

Figure S2. The timescale of haze events in Jinan city based on the statistic data.